# A Review of Robotic-Assisted Bronchoscopy Platforms in the Sampling of Peripheral Pulmonary Lesions

**DOI:** 10.3390/jcm10235678

**Published:** 2021-12-01

**Authors:** Michael Lu, Sridesh Nath, Roy W. Semaan

**Affiliations:** 1Division of Internal Medicine, University of Pittsburgh Medical Center, Pittsburgh, PA 15213, USA; lum@upmc.edu; 2Division of Pulmonary, Allergy and Critical Care Medicine, University of Pittsburgh Medical Center, Pittsburgh, PA 15213, USA; nathsg@upmc.edu

**Keywords:** robotic-assisted bronchoscopy, peripheral pulmonary lesions, lung cancer, Monarch, Ion

## Abstract

Robotic-assisted bronchoscopy is one of the newest additions to clinicians’ armamentarium for the biopsy of peripheral pulmonary lesions in light of the suboptimal yields and sensitivities of conventional bronchoscopic platforms. In this article, we review the existing literature pertaining to the feasibility as well as sensitivity of available robotic-assisted bronchoscopic platforms.

## 1. Introduction

The implementation of routine lung cancer screening guidelines, in addition to improvements in imaging modalities, has led to the increased detection of lung nodules. In recent decades, lung cancer has remained one of the leading causes of cancer-associated mortality [1], with tissue biopsy playing a crucial role in the diagnosis, staging, and genomic evaluation of lung cancers.

Two common modalities to sample lung lesions include image-guided trans-thoracic needle biopsy as well as endoscopic sampling. The majority of lung nodules are located in the lung periphery [2,3,4] and historically, trans-thoracic needle biopsy has been the preferred modality to sample peripheral pulmonary lesions (PPL) due to the technical challenges related to the endoscopic navigation of small peripheral airways. While trans-thoracic needle sampling has demonstrated impressive diagnostic yields [5], drawbacks include increased complication rates [6] as well as the need for subsequent endobronchial ultrasound (EBUS) procedures for mediastinal lymph node sampling for staging purposes should the sampled lung nodule(s) prove malignant on initial biopsy.

Conversely, endoscopic approaches have traditionally been limited to large and/or central lung nodules. While endoscopy allows for concurrent airway evaluation and mediastinal lymph node staging, inferior diagnostic yields and limited access to smaller PPLs certainly leave room for improvement.

Technologic innovations in recent years have led to improved imaging and navigation that has increased the accuracy, reach, and diagnostic yield of endoscopic sampling. The most recent of advances, robotic bronchoscopy, follows a national and global trend of the proliferation of robotic-assisted surgeries for thoracic [7,8] and extra-thoracic pathology [9,10] as well as oncologic staging such as axillary lymph node dissections [11]. The accompanying literature thus far demonstrates superiority or non-inferiority in numerous metrics including patient outcomes and medical cost compared to open or laparoscopic approaches. This review endeavors to outline the development of robotic bronchoscopy and the accompanying literature, demonstrating its advantages and disadvantages in comparison to more conventional bronchoscopic techniques.

## 2. Brief History

The flexible bronchoscope was developed by Dr. Shigeto Ikeda, a Japanese thoracic surgeon in the 1960s. Further revisions to the initial concept added tip angulation features in 1970 and video capabilities in 1987 [12]. The 1990s saw the introduction of radial probe endobronchial ultrasound (rEBUS), a rotating ultrasound transducer deployed from the working channel of flexible bronchoscopes. Hurter and Hanrath [13] demonstrated the feasibility of rEBUS in evaluating parenchymal and peribronchial lung lesions. However, years of rEBUS experience and numerous improvements in ultrasound technology have failed to show significant and consistent improvements in diagnostic yield as compared to conventional bronchoscopy with fluoroscopic guidance [14].

Navigational advances came in the form of electromagnetic navigational bronchoscopy (ENB), designed to guide operators to predetermined locations in the bronchial tree based on multi-planar computed tomography (CT) and real-time bronchoscope localization [15]. Although ENB was demonstrated to have a superior safety profile and greater success sampling PPLs as compared to flexible bronchoscopy, diagnostic yields vary greatly with contributing factors including nodule size, target location, bronchus sign, and concurrent use of rEBUS [16,17,18]. Furthermore, ENB is often reliant on preprocedural CT imaging that is vulnerable to CT-to-body divergence (CTBD), in which differences between the static images and the dynamic airways during intervention lead to discrepancies between the actual and expected location of target lesions. In response, new approaches such as real-time fluoroscopic evaluation [19] and the respiratory gating technology of the SPiN Thoracic Navigation System (Veran Medical Technologies, Inc., St. Louis, MO, USA) [20] have been developed and introduced to the market.

All said, sensitivities for CT-guided biopsy are as high as 92.1% [21] in the literature, with somewhat diminished yields (70–82%) when targeting smaller lesions [5]. Despite decades of experience with numerous endoscopic approaches, meta-analyses demonstrate sensitivities of rEBUS and ENB approaches at 70–75% [17,22,23,24], with some randomized-control trials demonstrating diagnostic yields for PPLs between 40 and 60% [14,18,25,26].

## 3. Robotic-Assisted Bronchoscopy

Numerous factors worsen the sensitivities of the bronchoscopic platform, including CTBD, misconstruing atelectasis for solid pulmonary nodules on rEBUS, as well as stiff biopsy instruments that impair the maneuverability of bronchoscopes. Due to these limitations, the robotic bronchoscopic platform was developed in an attempt to improve upon existing technology and optimizations.

Currently, two robotic platforms have gained approval from the Food and Drug Administration, namely the Monarch platform (Auris Health) and the Ion Endoluminal System (Intuitive Surgical). Key findings are discussed below and summarized in Table 1.

### 3.1. Monarch Robotic System

The Monarch system (Figure 1) consists of a bronchoscope system, cart, and tower. The bronchoscope system utilizes an outer sheath (6.0 mm diameter), an inner scope (4.4 mm diameter) that houses a camera as well as a 2.1 mm working channel. The outer sheath lends structural stability to the system and is generally driven to the lobar or segmental bronchi. The smaller inner scope is then navigated to the lesion through smaller peripheral airways.

Before the procedure, a CT is obtained with which virtual 3D reconstruction is generated. Potential pathways are mapped (manually or automatically) and uploaded to the robot. Patients lay within an electromagnetic field, allowing for the use of ENB for navigation purposes. The operator can control the bronchoscope with a handheld controller, not unlike that of a gaming controller. Once the target is reached, the location can be verified with fluoroscopy or rEBUS before the entire bronchoscope system is locked to maintain its static position while biopsies are obtained.

The Monarch system was first assessed in cadavers in the REACH study of 2018 [31]. Monarch was compared to conventional flexible bronchoscopy of a similar diameter (4.2 mm). Guidewires were advanced into segmental bronchi until pleura was reached, as confirmed by fluoroscopy, before either platform was advanced over the guidewire. The Monarch system was able to advance farther into distal airways compared to conventional bronchoscopy based on depth as well as airway generation (9th vs. 6th generations). Factors that were thought to have contributed include greater structural support, preventing instrument prolapse, as well as enhanced maneuverability, allowing for more apt negotiation of acute angulations.

The ACCESS Study [32] assessed the clinical implications of the greater range as demonstrated in the REACH study. Testing the Monarch’s ability to reach peripherally placed artificial targets in cadaveric subjects, the average size of target lesions was 20.4 mm with a mean distance of 16 mm from the pleural edge. Diagnostic yield was 94% (63/67) for transbronchial needle aspiration, increased to 97% (65/67) by the use of transbronchial forceps. Notably, the study demonstrated a diagnostic yield of 97% (47/48) with rEBUS, which was significantly higher than the yields of 30–40% documented in the literature. This difference was thought to be due to improved articulation and control of the distal end of the bronchoscope, with the caveat that the cadaveric tissue/models used in the study may not reflect the dynamic changes made by live human tissue in the periprocedural setting.

The first human feasibility study was performed by Rojas-Solano et al. in 2018 [33]. The study included 15 patients with suspicious pulmonary lesions with bronchus sign confirmed on CT (12 peripheral and 3 central lesions) with an average size of 26 mm (10–63 mm). Of the 15 lesions, 14 (93%) were able to be sampled successfully. No complications such as pneumothorax or postprocedural bleeding were reported.

A retrospective post-market study by Chaddha et al. analyzed 167 lesions in 165 patients [27]. The average lesion size was 25 mm (10–40 mm) with 71.3% of lesions smaller than 30 mm. The majority (70.7%) of lesions were located in the outer third of the lung and 63.5% of lesions had a bronchus sign. Operators successfully navigated to 88.6% (148/167) of lesions as confirmed by rEBUS. Tissue biopsy was successfully obtained in 97.6% (161/165) of patients, with an overall diagnostic yield of 69.1–77% as all biopsy-proven cases of inflammation without follow-up data (13 cases) were deemed non-diagnostic. Lesion size, density, lobar location, and centrality did not affect diagnostic yield. Pneumothorax was found in 3.6% (6/165) of patients with 2.4% (4/165) of patients requiring tube thoracostomy. Significant airway bleeding was noted in 2.4% (4/165) of cases.

The BENEFIT study was the first prospective multi-centered study with the Monarch system, consisting of 54 patients across 5 centers [28]. Median lesion size was 23 mm (15–29 mm) with 78% of lesions being smaller than 30 mm. Following navigation to the target lesion, localization was able to be confirmed by rEBUS in 96.2% (51/53) of cases. The diagnostic yield in this study was 74.1% (40/54). Specifically, the yield was 80.6% (25/31) for PPLs with a concentric view and 70% for PPLs with an eccentric view. The safety profile was comparable to those of conventional bronchoscopy [25], with post-procedural pneumothorax found in 3.7% (2/54) of cases, of which 1.9% (1/54) of cases needed tube thoracostomy. A median procedure time of 51 min was also comparable to early experiences with ENB.

### 3.2. Ion Endoluminal

The Ion system (Figure 2) has a bronchoscope with a single ultra-thin scope with a 3.5 mm outer diameter and a 2.0 mm working channel with a vision probe that is passed through the working channel.

Similar to Monarch, the Ion system uses a pre-procedure CT Chest that is uploaded to their PlanPoint program for 3D reconstruction and target pathway planning. Rather than ENB, Ion uses shape-sensing catheter technology as its modality for localization. The operator is able to navigate the bronchoscope using a trackball controller. At the start of the procedure, the vision probe is fed through the shape-sensing catheter during registration and navigation to target lesions. Once the target is reached, localization can be similarly confirmed with fluoroscopy or rEBUS before the bronchoscope is fixed in place. Notably, the vision probe needs to be retracted from the catheter prior to the passing of biopsy equipment and tissue acquisition under fluoroscopy visualization.

The first feasibility study for Ion in human subjects included 29 consecutive cases in a single center [29]. Average lesion size was 12 mm (10–30 mm) with 58.6% of lesions having a bronchus sign present. Lesions were able to be reached 96.6% (28/29) of the time, as confirmed by rEBUS with an overall diagnostic yield of 79.3% (23/29).

In a cadaveric comparative study of the Ion system vs. ENB vs. rEBUS [34], implanted targets were placed in five cadaveric models with a mean size of 16.5 mm; 80% of targets were placed within the outer one-third of the lung. The study found that localization resulting in successful needle-in-nodule (as demonstrated on CT) was seen in 80% (16/20) of cases using Ion as compared to 45% (9/20) with ENB and 25% (5/20) with rEBUS. Even in instances where the lesion was missed, the median needle-to-miss distance was 4 mm in the Ion platform vs. 7 mm with ENB and 13 mm with rEBUS. Similar to the cadaveric studies in the Monarch system, limitations to the study include dissimilarities in the tissue characteristics of cadavers and implanted targets compared to live human subjects and peripheral lung lesions.

Recently published was a retrospective observational single-center study at a high-volume academic institution [30]. This study included 130 patients with 159 lesions. Median lesion size was 18 mm with 39% (62/159) of lesions located in the lung periphery. Successful navigation was achieved in 98.7% (157/159) of lesions with an overall diagnostic yield of 81.7%. Notable merits of the Ion described include 69.5% diagnostic yield in lesions < 20 mm, 71.1% yield in lesions where bronchus sign was absent, as well as 70.9% yield in lesions located in the periphery, all figures that are greater than those previously described in the literature. Complication rate was reported at 3% (4/130) with 1.5% (2/130) of patients developing post-procedural pneumothorax requiring tube thoracostomy. Other adverse events included a single case each of post-procedural aspiration pneumonia and hypoxemia that were successfully managed with conservative therapy.

## 4. Further Directions

In addition to the localization of peripheral lung lesions, the robotic systems may offer another modality with which to apply therapeutic interventions for the minimally invasive treatment of lung cancer in the near future. Studies will soon start to examine the use of microwave ablation of early-stage lung cancer in patients who are not surgical candidates as well as local injection therapies. These therapies will also take advantage of cone beam CT technology to ensure appropriate positioning of the catheters. Other therapeutic modalities such as photodynamic therapy, radiofrequency ablation, and YAG laser may also be used in the future as well. Finally, the application of dyes and fiducial markers has also been recently performed for improved lesion localization prior to surgical resection as well as stereotactic radiosurgery. That said, the performance of endoluminal procedures with robotic bronchoscopic systems has yet to be documented in the literature and we await future studies to compare efficacy to existing technologies and procedures.

## 5. Conclusions

The perceived advantages of the two robotic platforms hinge upon improved tip maneuverability and system stability to enable operators to more easily traverse into peripheral lung fields and obtain biopsy samples of PPLs. Notable drawbacks of robotic bronchoscopy are similar to those noted at the early stages of adoption of other robotic-assisted surgeries, such as increased unit cost as compared to traditional approaches and additional training to OR staff to support robotic platforms. Furthermore, the need to dock/un-dock platforms is thought to lengthen procedure times and response latency to peri-procedural complications. That said, what has been demonstrated in the surgical literature is that these disadvantages can often be ameliorated with operator/staff fluency, seen especially at high-volume centers [9,35].

The existing literature, by way of single and multi-centered feasibility studies, thus far has demonstrated promising results with a comparable safety profile. Studies currently in the pipeline include the multi-centered, prospective, observational TARGET trial designed to assess the clinical safety and diagnostic accuracy of the Monarch platform (NCT04182815), as well as the prospective, multi-center PRECIsE trial, evaluating the clinical utility and performance of the Ion System (NCT03893539).

As the new technology makes its way into hospitals and more single-center experiences are reported [30,36], the body of literature to characterize the performance of these platforms grows. With broader adoption and higher-powered literature on the horizon, it remains to be seen if these technologic advantages will bear improved sensitivities in the biopsy of peripheral pulmonary lesions.

## Figures and Tables

**Figure 1 jcm-10-05678-f001:**
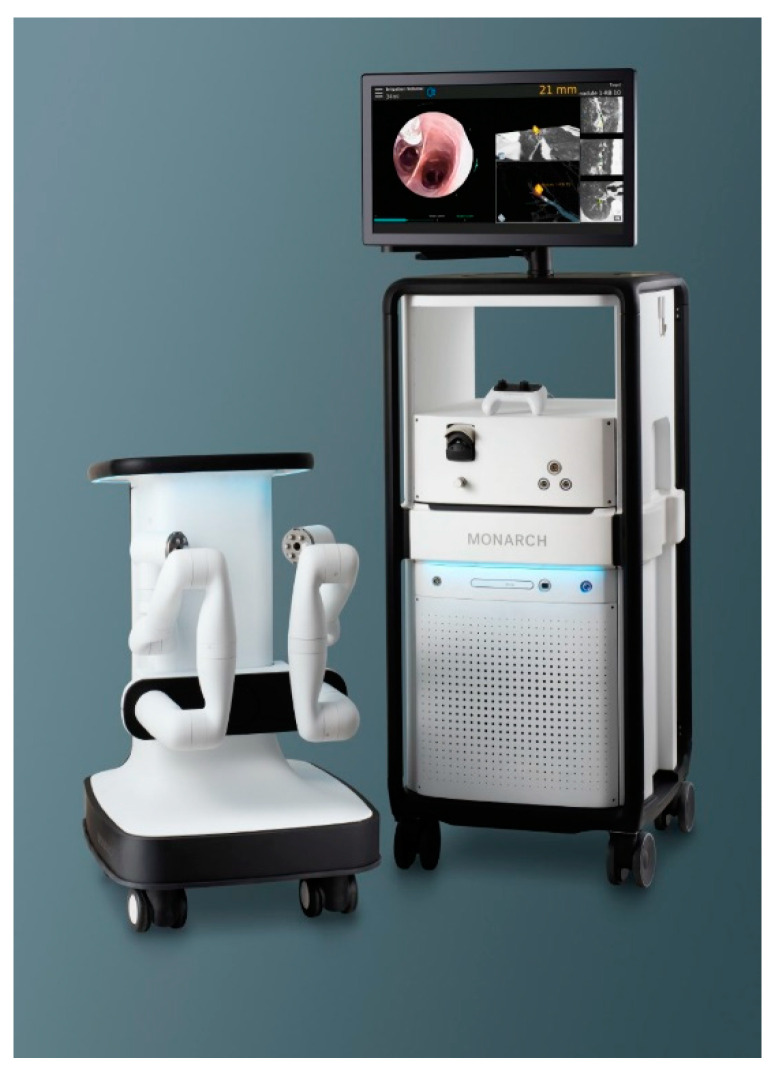
Monarch^®^ Robotic-Assisted Bronchoscopy Platform by Auris Health.

**Figure 2 jcm-10-05678-f002:**
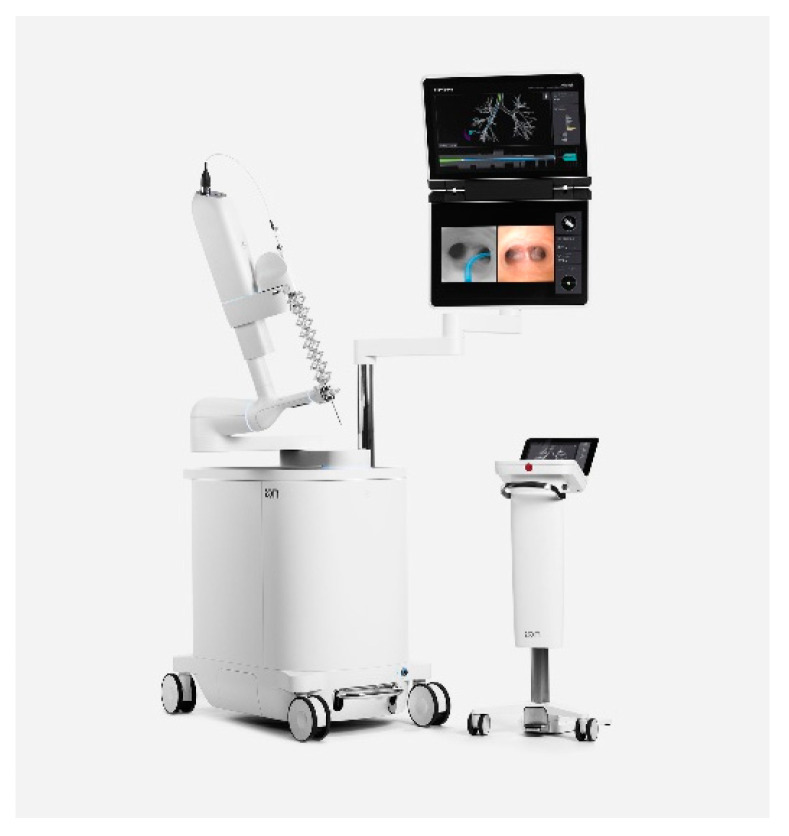
Ion™ Endoluminal System by Intuitive Surgical.

**Table 1 jcm-10-05678-t001:** Diagnostic sensitivity and complication rates of different modalities.

	Lesion Size (mm)	Diagnostic Sensitivity	Pneumothorax Rate
Guided Bronchoscopy * [18]	NS	70%	1.5%
CT-guided Transthoracic Needle Biopsy [21]	NS	92.1%	20.5%
Robotic Platform Studies			
Chaddha et al. [27] (*n* = 165)	25.0 ± 15.0	69.1–77%	3.6%
Chen et al. (BENEFIT study) [28] (*n* = 54)	23.2 ± 10.8	74.1%	3.7%
Fielding et al. [29] (*n* = 29)	15.3 ± 4.8	79.3%	0%
Kalchiem-Dekel et al. [30] (*n* = 130)	18.0 ± 9.0	79.8%	1.5%

*n* = number of lesions biopsied (except Chaddha et al. and Kalchiem-Dekel et al. where *n* = number of cases); NS = not specified; * = guided bronchoscopy includes electromagnetic navigation bronchoscopy (ENB), virtual bronchoscopy (VB), radial endobronchial ultrasound (rEBUS), ultrathin bronchoscope, and guide sheath.

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
