# Peer review of "A Review of Robotic-Assisted Bronchoscopy Platforms in the Sampling of Peripheral Pulmonary Lesions"

_jcm, 2021, doi:10.3390/jcm10235678_

Round 1

Reviewer 1 Report

Dear authors:

In the manuscript by Lu et al., the authors show the advantages of robotic-assisted bronchoscopy platforms in the sampling of peripheral pulmonary lesions. I would just give a few suggestions.

1. Please modify and improve the introduction and add more citations. Please introduce more about applications of Surgical Robot in other fields in clinical medicine, in order to highlight the popularity of robots and robot platforms is the general trend, for instance, introduce the applications of Da Vinci Robots for axillary lymph node dissection (dVALND) in breast cancers and so on(please cite, Chen K et al. Efficacy of da Vinci robot-assisted lymph node surgery than conventional axillary lymph node dissection in breast cancer - A comparative study. Int J Med Robot. 2021 Jul 16:e2307. doi: 10.1002/rcs.2307.) 

2. Please further describe and give more information about the advantages and disadvantages of the robot-assisted platform compared with the traditional method.

Best,

Author Response

Thank you for the opportunity to submit a revision for manuscript titled "A review of robotic-assisted bronchoscopy platforms in the sampling of peripheral pulmonary lesions." for publication consideration in JCM. The constructive comments by the reviewers have provided us with an opportunity to greatly strengthen and focus the manuscript. In response to the review comments:

  1. Please modify and improve the introduction and add more citations. Please introduce more about applications of Surgical Robot in other fields in clinical medicine, in order to highlight the popularity of robots and robot platforms is the general trend, for instance, introduce the applications of Da Vinci Robots for axillary lymph node dissection (dVALND) in breast cancers and so on(please cite, Chen K et al. Efficacy of da Vinci robot-assisted lymph node surgery than conventional axillary lymph node dissection in breast cancer - A comparative study. Int J Med Robot. 2021 Jul 16:e2307. doi: 10.1002/rcs.2307.)

- Thank you for bringing up this point.  In response, we have added 6 references to the introduction discussing the expanding field of robotic surgery with thoracic and extra thoracic examples, including the aforementioned publication in-press on breast lymph node dissections.

2. Please further describe and give more information about the advantages and disadvantages of the robot-assisted platform compared with the traditional method.

- The authors agree further details on advantages/disadvantages will certainly be helpful to the reader.  We have expanded upon these in the discussion section from lines 208 to 215, touching upon common drawbacks noted during early stages of adoption of most robotic surgical platforms. 

The final manuscript has been approved by all the authors, and we have taken due care to ensure the integrity of the work.  The work is not currently under consideration at any other journal. Thank you for your consideration. I look forward to hearing from you.  

Reviewer 2 Report

The Authors presents this review about robotic-assisted bronchoscopy platforms in the sampling of peripheral pulmonary lesions.The perceived advantages of the two robotic platforms hinge upon improved ma neuverability and stability to enable operators to more easily traverse into peripheral lung fields and localize PPLs. I congratulate with the authors for this review well written and with high scientific impact.

Author Response

Thank you for the opportunity to submit a revision for manuscript titled "A review of robotic-assisted bronchoscopy platforms in the sampling of peripheral pulmonary lesions." for publication consideration in JCM. The constructive comments by the reviewers have provided us with an opportunity to greatly strengthen and focus the manuscript.

The final manuscript has been approved by all the authors, and we have taken due care to ensure the integrity of the work.  The work is not currently under consideration at any other journal. Thank you for your consideration. I look forward to hearing from you.  

Round 2

Reviewer 1 Report

Authors made correction according to my previous suggestions. Strongly recommend for publishing.

Best

Author Response

Thank you again for the recommendations.